# Stabilizing GAN Training with Multiple Random Projections

## Abstract

Training generative adversarial networks is unstable in high-dimensions as the true data distribution tends to be concentrated in a small fraction of the ambient space. The discriminator is then quickly able to classify nearly all generated samples as fake, leaving the generator without meaningful gradients and causing it to deteriorate after a point in training. In this work, we propose training a single generator simultaneously against an array of discriminators, each of which looks at a different random low-dimensional projection of the data. Individual discriminators, now provided with restricted views of the input, are unable to reject generated samples perfectly and continue to provide meaningful gradients to the generator throughout training. Meanwhile, the generator learns to produce samples consistent with the full data distribution to satisfy all discriminators simultaneously. We demonstrate the practical utility of this approach experimentally, and show that it is able to produce image samples with higher quality than traditional training with a single discriminator.

## 1 Introduction

Generative adversarial networks (GANs), introduced by Goodfellow et al. (2014), endow neural networks with the ability to express distributional outputs. The framework includes a generator network that is tasked with producing samples from some target distribution, given as input a (typically low dimensional) noise vector drawn from a simple known distribution, and possibly conditional side information. The generator learns to generate such samples, not by directly looking at the data, but through adversarial training with a discriminator network that seeks to differentiate real data from those generated by the generator. To satisfy the objective of "fooling" the discriminator, the generator eventually learns to produce samples with statistics that match those of real data.

In regression tasks where the true output is ambiguous, GANs provide a means to simply produce an output that is plausible (with a single sample), or to explicitly model that ambiguity (through multiple samples). In the latter case, they provide an attractive alternative to fitting distributions to parametric forms during training, and employing expensive sampling techniques at the test time. In particular, conditional variants of GANs have shown to be useful for tasks such as in-painting (Denton et al., 2016), and super-resolution (Ledig et al., 2016). Recently, Isola et al. (2016) demonstrated that GANs can be used to produce plausible mappings between a variety of domains—including sketches and photographs, maps and aerial views, segmentation masks and images, *etc.* GANs have also found uses as a means of un-supervised learning, with latent noise vectors and hidden-layer activations of the discriminators proving to be useful features for various tasks (Denton et al., 2016; Chen et al., 2016; Radford et al., 2016).

Despite their success, training GANs to generate high-dimensional data (such as large images) is challenging. Adversarial training between the generator and discriminator involves optimizing a min-max objective. This is typically carried out by gradient-based updates to both networks, and the generator is prone to divergence and mode-collapse as the discriminator begins to successfully distinguish real data from generated samples with high confidence. Researchers have tried to address this instability and train better generators

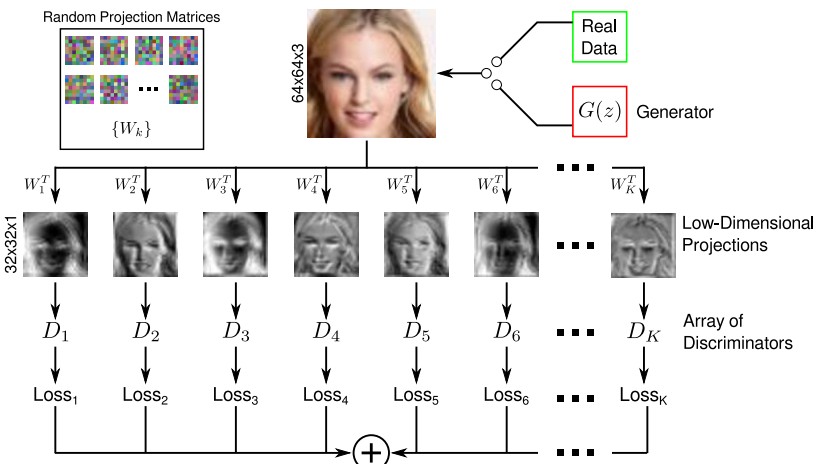

Figure 1: Overview of our approach. We train a single generator against an array of discriminators, each of which receives lower-dimensional projections—chosen randomly prior to training—as input. Individually, these discriminators are unable to perfectly separate real and generated samples, and thus provide stable gradients to the generator throughout training. In turn, by trying to fool all the discriminators simultaneously, the generator learns to match the true full data distribution.

through several techniques. Denton et al. (2015) proposed generating an image by explicitly factorizing the task into a sequence of conditional generations of levels of a Laplacian pyramid, while Radford et al. (2016) demonstrated that specific architecture choices and parameter settings led to higher-quality samples. Other techniques include providing additional supervision (Salimans et al., 2016), adding noise to the discriminator input (Arjovsky & Bottou, 2016), as well as modifying or adding regularization to the training objective functions (Nowozin et al., 2016; Arjovsky et al., 2017; Zhao et al., 2016).

We propose a different approach to address this instability, where the generator is trained against an array of discriminators, each of which looks at a different, randomly-chosen, low-dimensional projection of the data. Each discriminator is unable to perfectly separate real and generated samples since it only gets a partial view of these samples. At the same time, to satisfy its objective of fooling *all* discriminators, the generator learns to match the true full data distribution. We describe a realization of this approach for training image generators, and discuss the intuition behind why we expect this training to be stable—i.e., the gradients to the generator to be meaningful throughout training—and consistent—i.e., for the generator to learn to match the full real data distribution. Despite its simplicity, we find this approach to be surprisingly effective in practice. We demonstrate this efficacy by using it to train generators on standard image datasets, and find that these produce higher-quality samples than generators trained against a single discriminator.

## 1.1 Related Work

Researchers have explored several approaches to improve the stability of GANs for training on higher dimensional images. Instead of optimizing Jensen-Shannon divergence as suggested in the original GAN framework, Energy based GAN (Zhao et al., 2016) and Wasserstein GAN (Arjovsky et al., 2017) show improvement in the stability by optimizing total variation distance and Earth mover distance respectively, together with regularizing the discriminator to limit the discriminator capacity. Nowozin et al. (2016) further extended GAN training to any choice of $f$-divergence as objective. Salimans et al. (2016) proposed several heuristics to improve the stability of training. These include modifying the objective function, virtual batch-normalization, historical averaging of parameters, semi-supervised learning, *etc.*. All of these methods are designed to improve the quality of gradients, and provide additional supervision to the generator. They are therefore complementary to, and can likely be used in combination with, our approach.

Note that prior methods have involved training ensembles of GANs (Wang et al., 2016), or ensembles of discriminators (Durugkar et al., 2016). However, their goal is different from ours. In our framework, each discriminator is shown only a low-dimensional projection of the data, with the goal of preventing it from being able to perfectly reject generated samples. Crucially, we do not combine the outputs of discriminators directly, but rather compute losses on individual discriminators and average these losses.

Indeed, the idea of combining multiple simple classifiers, *e.g.,* with boosting (Freund et al., 1996), or mixture-of-experts Jacobs (1995), has a rich history in machine learning, including recent work where the simple classifiers act on random lower-dimensional projections of their inputs (Cannings & Samworth, 2017) (additionally benefiting from the regularization effect of such projections (Durrant & Kabán, 2015)). While these combinations have been aimed at achieving high classification accuracy (i.e., achieve accurate discrimination), our objective is to maintain a flow of gradients from individual discriminators to the generator. It is also worth noting here the work of Bonneel et al. (2015), who also use low-dimensional projections to deal with high-dimensional probability distributions. They use such projections to efficiently compute Wasserstein distances and optimal transport between two distributions, while we seek to enable stable GAN training in high dimensions.

## 2    Training with Multiple Random Projections

GANs (Goodfellow et al., 2014) traditionally comprise of a generator $G$ that learns to generate samples from a data distribution $\mathbb{P}_x$, through adversarial training against a single discriminator $D$ as:

$$\min_G \ \max_D \ V(D, G) = \mathbb{E}_{x \sim \mathbb{P}_x}[\log D(x)] + \mathbb{E}_{z \sim \mathbb{P}_z}[\log(1 - D(G(z)))], \qquad (1)$$

for $x \in \mathbb{R}^d$, $\mathbb{P}_x : \mathbb{R}^d \to \mathbb{R}_+, \int \mathbb{P}_x = 1$, and with $\mathbb{P}_z$ a fixed distribution (typically uniform or Gaussian) for noise vectors $z \in \mathbb{R}^{d'}, d' \ll d$. The optimization in (1) is carried out using stochastic gradient descent (SGD), with alternating updates to the generator and the discriminator. While this approach works surprisingly well, instability is common during training, especially when generating high-dimensional data.

Theoretically, the optimal stationary point for (1) is one where the generator matches the data distribution perfectly, and the discriminator is forced to always output 0.5 (Goodfellow et al., 2014). But usually in practice, the discriminator tends to "win" and the cost in (1) saturates at a high value. While the loss keeps increasing, the gradients received by the generator are informative during the early stages of training. However, once the loss reaches a high enough value, the gradients are dominated by noise, at which point generator quality stops improving and in fact begins to deteriorate. Training must therefore be stopped early by manually inspecting sample quality, and this caps the number of iterations over which the generator is able to improve.

Arjovsky & Bottou (2016) discuss one possible source of this instability, suggesting that it is because natural data distributions $\mathbb{P}_x$ often have very limited support in the ambient domain of $x$ and $G(z)$. Then, the generator $G$ is unable to learn quickly enough to generate samples from distributions that have significant overlap with this support. This in turn makes it easier for the discriminator $D$ to perfectly separate the generator's samples, at which point the latter is left without useful gradients for further training.

We propose to ameliorate this problem by training a *single* generator against an *array* of discriminators. Each discriminator operates on a different low-dimensional linear projection, that is set randomly prior to training. Formally, we train a generator $G$ against multiple discriminators $\{D_k\}_{k=1}^K$ as:

$$\min_G \ \max_{\{D_k\}} \ \sum_{i=k}^K V(D_k, G) = \sum_{i=k}^K \mathbb{E}_{x \sim \mathbb{P}_x}[\log D_k(W_k^T x)] + \mathbb{E}_{z \sim \mathbb{P}_z}[\log(1 - D_k(W_k^T G(z)))], \quad (2)$$

where $W_k, k \in \{1, \cdots, K\}$ is a randomly chosen matrix in $\mathbb{R}^{d \times m}$ with $m < d$. Therefore, instead of a single discriminator looking at the full input (be it real or fake) in (1), each of

the $K$ discriminators in (2) sees a different low-dimensional projection. Each discriminator tries to maximize its accuracy at detecting generated samples from its own projected version, while the generator tries to ensure that each sample it generates simultaneously fools all discriminators for all projected versions of that sample.

When the true data $X$ and generated samples $G(z)$ are images, prior work (Radford et al., 2016) has shown that it is key that both the generator and discriminator have convolutional architectures. While individual discriminators in our framework see projected inputs that are lower-dimensional than the full image, their dimensionality is still large enough (a very small $m$ would require a large number of discriminators) to make it hard to train discriminators with only fully-connected layers. Therefore, it is desirable to employ convolutional architectures for each discriminator. To do so, the projection matrices $W_k^T$ must be chosen to produce "image-like" data. Accordingly, we use strided convolutions with random filters to embody the projections $W_k^T$ (as illustrated in Fig. 1). The elements of these convolution filters are drawn i.i.d. from a Gaussian distribution, and the filters are then scaled to have unit $\ell_2$ norm. To promote more "mixing" of the input coordinates, we choose filter sizes that are larger than the stride (e.g., we use $8 \times 8$ filters when using a stride of 2). While still random, this essentially imposes a block-Toeplitz structure on $W_k^T$.

## 3 Motivation

In this section, we motivate our approach, and provide the reader with some intuition for why we *expect* our approach to improve the stability of training while maintaining consistency.

**Stability.** Each discriminators $D_k$ in our framework in (2) works with low dimensional projections of the data and can do no better than the traditional full discriminator $D$ in (1). Let $y \in \mathbb{R}^d$ and $l \in \{0, 1\}$ denote the input and ideal output of the original discriminator $D$, where $l = 1$ if $y$ is real, and 0 if generated. However, each $D_k$ is provided only a lower dimensional projection $W_k^T y$. Since $W_k^T y$ is a deterministic and non-invertible function of $y$, it can contain no more information regarding $l$ than $Y$ itself, i.e., $I(l; Y) \geq I(l; W_k^T Y)$, where $I(x; y)$ denotes the mutual information between the random variables $x$ and $y$. In other words, taking a low-dimensional projection introduces an information bottleneck which interferes with the ability of the discriminator to determine the real/fake label $l$.

While strictly possible, it is intuitively unlikely that all the information required for classification is present in the $W_k^T y$ (for all, or even most of the different $W_k^T$), especially in the adversarial setting. We already know that GAN training is far more stable in lower dimensions, and we expect that each of the discriminators $D_k$ will perform similarly to a traditional discriminator training on lower (i.e., $m-$) dimensional data. Moreover, Arjovsky & Bottou (2016) suggest that instability in higher-dimensions is caused by the true distribution $\mathbb{P}_x$ being concentrated in a small fraction of the ambient space—i.e., the support of $\mathbb{P}_x$ occupies a low volume relative to the range of possible values of $x$. Again, we can intuitively expect this to be ameliorated by performing a random lower-dimensional projection (and provide analysis for a simplistic $\mathbb{P}_x$ in the supplementary).

When the discriminators $D_k$ are not perfectly saturated, $D_k(W_k^T G(z))$ will have some variation in the neighborhood of a generated sample $G(z)$, which can provide meaningful gradients to the generator. But note that gradients from each individual discriminator $D_k$ to $G(z)$ will lie entirely in the lower dimensional sub-space spanned by $W_k^T$.

**Consistency.** While impeding the discriminator's ability benefits stability, it could also have trivially been achieved by other means—e.g., by severely limiting its architecture or excessive regularization. However, these would also limit the ability of the discriminator to encode the true data distribution and pass it on to the generator. We begin by considering the following modified version of Theorem 1 in Goodfellow et al. (2014):

**Theorem 3.1.** *Let $\mathbb{P}_g$ denote the distribution of the generator outputs $G(z)$, where $z \sim \mathbb{P}_z$, and let $\mathbb{P}_{W_k^T g}$ be the marginals of $\mathbb{P}_g$ along $W_k$. For fixed $G$, the optimal $\{D_k\}$ are given by*

$$D_k(y) = \mathbb{P}_{W_k^T x}(y) / \left( \mathbb{P}_{W_k^T x}(y) + \mathbb{P}_{W_k^T g}(y) \right), \qquad (3)$$

*for all $k \in \{1, \cdots, K\}$. The optimal $G$ under (2) is achieved iff $\mathbb{P}_{W_k^T x} = \mathbb{P}_{W_k^T g}$, $\forall k$.*

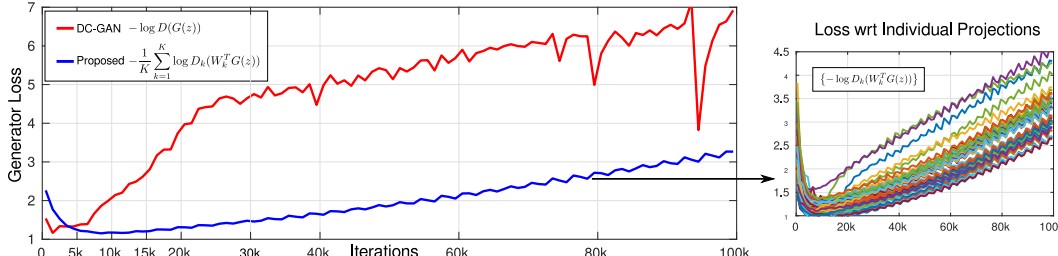

Figure 2: Training Stability. We plot the evolution of the "generator loss" across training—against a traditional single discriminator (DC-GAN), and the average and individual losses against multiple discriminators ($K = 48$) in our setting. For the traditional single discriminator, this loss rises quickly to high value, indicating that the discriminator saturates to rejecting generated samples with very high confidence. In contrast, the loss in our case remains lower, allowing our discriminators to provide meaningful gradients to the generator.

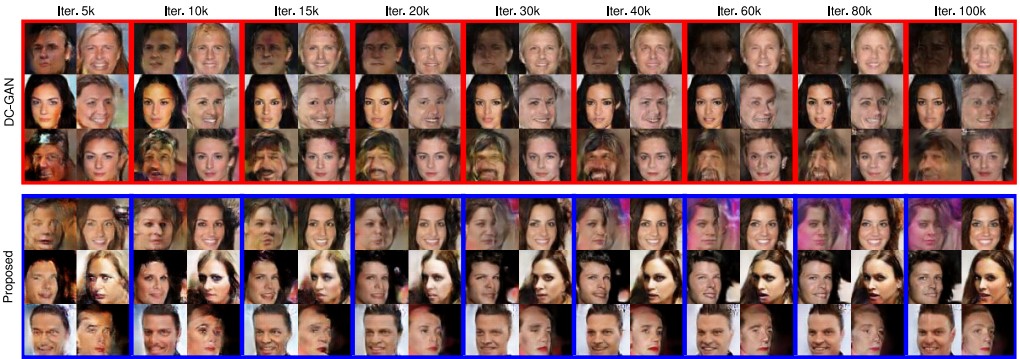

Figure 3: Evolution of sample quality across training iterations. With our approach, the generator improves the visual quality of its samples quickly and throughout training. Meanwhile, the generator trained in the traditional setting with a single discriminator shows slower improvement, and indeed quality begins to deteriorate after a point.

This result, proved in the supplementary, implies that under ideal conditions the generator will produce samples from a distribution whose *marginals* along each of the projections $W_k$ match those of the true distribution. Thus, each discriminator adds an additional constraint on the generator, forcing it to match $\mathbb{P}_x$ along a different marginal. As we show in the supplementary, matching along a sufficiently high number of such marginals—under smoothness assumptions on the true and generator distributions $\mathbb{P}_x$ and $\mathbb{P}_g$—guarantees that the full joint distributions will be close (note it isn't sufficient for the set of projection matrices $\{W_k\}$ to simply span $\mathbb{R}^d$ for this). Therefore, even though viewing the data through random projections limits the ability of individual discriminators, with enough discriminators acting in concert, the generator learns to match the full joint distribution of real data in $\mathbb{R}^d$.

## 4 EXPERIMENTAL RESULTS

We now evaluate our approach with experiments comparing it to generators trained against a single traditional discriminator, and demonstrate that it leads to higher stability during training, and ultimately yields generators that produce higher quality samples.

**Dataset and Architectures.** For evaluation, we primarily use the dataset of celebrity faces collected by Liu et al. (2015)—we use the cropped and aligned $64 \times 64$-size version of the images—and the DC-GAN (Radford et al., 2016) architectures for the generator and discriminator. We make two minor modifications to the DC-GAN implementation that we find empirically to yield improved results (for both the standard single discriminator setting, as well as our multiple discriminator setting). First, we use different batches of generated

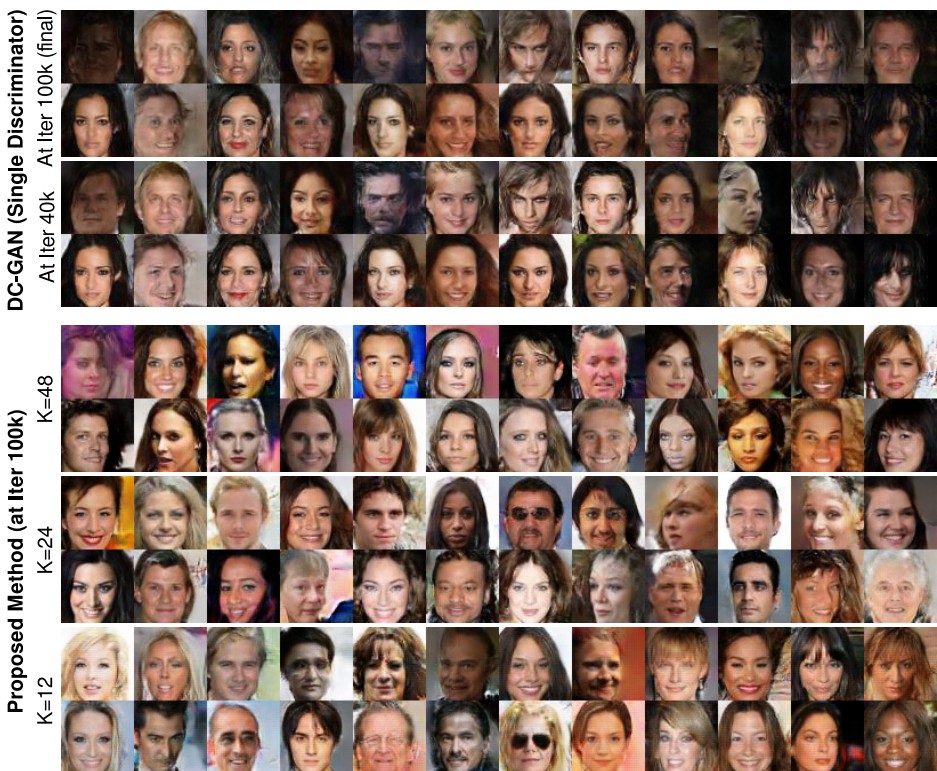

Figure 4: Random sets of generated samples from traditional DC-GAN and the proposed framework. For DC-GAN, we show results from the model both at 40k iterations (when the samples are qualitatively the best) and at the end of training (100k iterations). For our setting, we show samples from the end of training for generator models trained with $K = 12, 24, 48$ projections. Our generator produces qualitatively better samples, with finer detail and fewer distortions. Quality is worse with subtle high-frequency noise when $K$ is smaller, but these decrease with increasing $K$ to 24 and 48.

images to update the discriminator and generator—this increases computation per iteration, but yields better quality results. Second, we employ batch normalization in the generator but *not* in the discriminator (the original implementation normalized real and generated batches separately, which we found yielded poorer generators).

For our approach, we train a generator against $K$ discriminators, each operating on a different single-channel $32 \times 32$ projected version of the input, *i.e.*, $d/m = 12$. The projected images are generated through convolution with $8 \times 8$ filters and a stride of two. The filters are generated randomly and kept constant throughout training. We compare this to the standard DC-GAN setting of a single discriminator that looks at the full-resolution $64 \times 64$ color image. We use identical generator architectures in both settings—that map a 100 dimensional uniformly distributed noise vector to a full resolution image. The discriminators also have similar architectures—but each of the discriminator in our setting has one less layer as it operates on a lower resolution input (we map the number of channels in the first layer in our setting to those of the second layer of the full-resolution single-discriminator, thus matching the size of the final feature vector used for classification). As is standard practice, we compute generator gradients in both settings with respect to minimizing the "incorrect" classification loss of the discriminator—in our setting, this is given by $-\frac{1}{K}\sum_k \log D_k(G(z))$. As suggested in (Radford et al., 2016), we use Adam (Kingma & Ba, 2014) with learning rate $2 \times 10^{-4}$, $\beta_1 = 0.5$, and a batch size of 64.

**Stability.** We begin by analyzing the evolution of generators in both settings through training. Figure 2 shows the generator training loss for traditional DC-GAN with a single discriminator, and compares it the proposed framework with $K = 48$ discriminators. In both

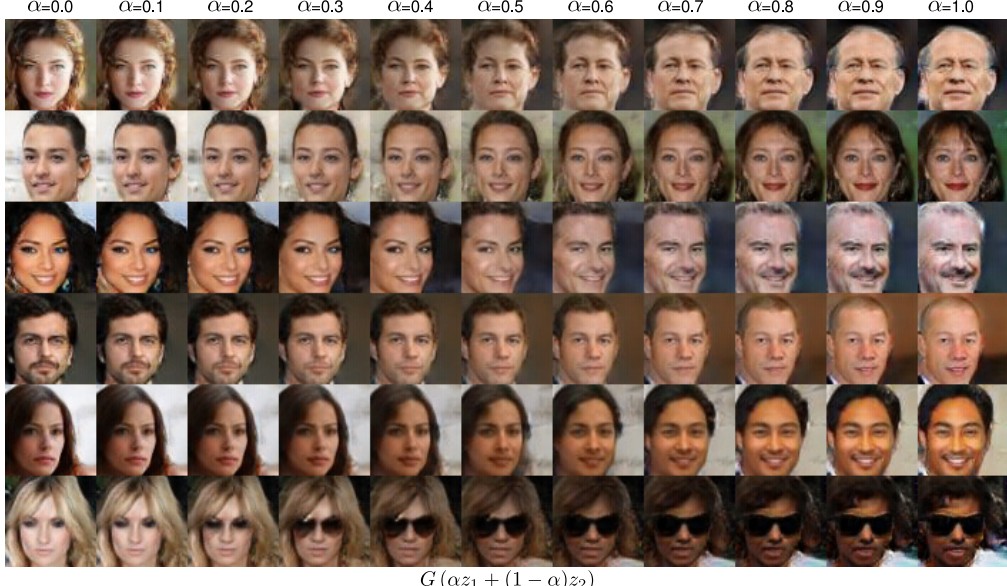

$$G(\alpha z_1 + (1 - \alpha)z_2)$$

Figure 5: Interpolating in latent space. For selected pairs of generated faces (with $K = 48$), we generate samples using different convex combinations of their corresponding noise vectors. Every combination generates a plausible face, and these appear to smoothly interpolate between various facial attributes—age, gender, expression, hair, *etc.* Note that the $\alpha = 0.5$ sample always corresponds to an individual clearly distinct from the original pair.

settings, the generator losses increase through much of training after decreasing in the initial iterations (*i.e.,* the discriminators eventually "win"). However, DC-GAN's generator loss rises quickly and remains higher than ours in absolute terms throughout training. Figure 3 includes examples of generated samples from both generators across iterations (from the same noise vectors). We observe that DC-GAN's samples improve mostly in the initial iterations while the training loss is still low, in line with our intuition that generator gradients become less informative as discriminators get stronger. Indeed, the quality of samples from traditional DC-GAN actually begins to deteriorate after around 40k iterations. In contrast, the generator trained in our framework improves continually throughout training.

**Consistency.** Beyond stability, Fig. 3 also demonstrates the consistency of our framework. While the average loss in our framework is lower, we see this does not impede our generator's ability to learn the data distribution quickly as it collates feedback from multiple discriminators. Indeed, our generator produces higher-quality samples than traditional DC-GAN even in early iterations. Figure 4 includes a larger number of (random) samples from generators trained with traditional DC-GAN and our setting. For DC-GAN, we include samples from both roughly the end (100k iterations) of training, as well as from roughly the middle (40k iterations) where the sample quality are approximately the best. For our approach, we show results from training with different numbers of discriminators with $K = 12, 24$, and 48—selecting the generator models from the end of training for all. We see that our generators produce face images with higher-quality detail and far fewer distortions than traditional DC-GAN. We also note the effect of the number of discriminators on sample quality. Specifically, we find that setting $K$ to be equal only to the projection ratio $d/m = 12$ leads to subtle high-frequency noise in the generator samples, suggesting these many projections do not sufficiently constrain the generator to learn the full data distribution. Increasing $K$ diminishes these artifacts, and $K = 24$ and 48 both yield similar, high-quality samples.

**Training Time.** Note that the improved generator comes at the expense of increased computation during training. Traditional DC-GAN with a single discriminator takes only $0.6s$ per training iteration (on an NVIDIA Titan X), but this goes up to $3.2s$ for $K = 12$, $5.8s$ for $K = 24$, and $11.2s$ for $K = 48$ in our framework. Note that once trained, all generators produce samples at the same speed as they have identical architectures.

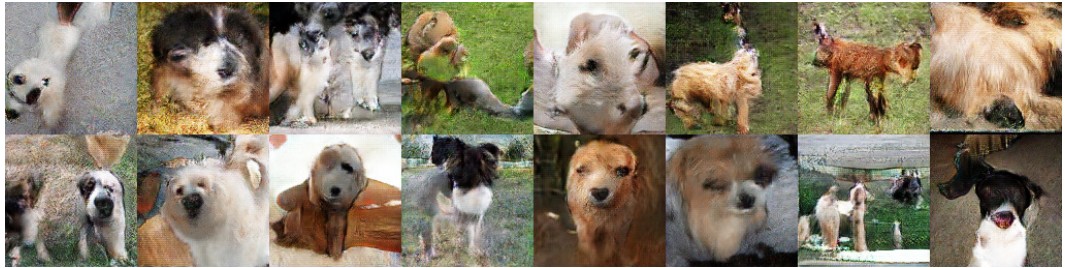

Figure 6: Examples from training on canine images from Imagenet. We show manually selected examples of $128 \times 128$ images produced by a generator trained on various canine classes from Imagenet. Although not globally plausible, the samples contain realistic low-level textures, and reproduce rough high-level composition.

**Latent Embedding.** Next, we explore the quality of the embedding induced by our generator ($K = 48$) in the latent space of noise vectors $z$. We consider selected pairs of randomly generated faces, and show samples generated by linear interpolation between their corresponding noise vectors in Fig. 5. Each of these generated samples is also a plausible face image, which confirms that our generator is not simply memorizing training samples, and that it is densely packing the latent space with face images. We also find that the generated samples smoothly interpolate between semantically meaningful face attributes—gender, age, hair-style, expression, and so on. Note that in all rows, the sample for $\alpha = 0.5$ appears to be a clearly different individual than the ones represented by $\alpha = 0$ and $\alpha = 1$.

**Results on Imagenet-Canines.** Finally, we show results on training a generator on a subset of the Imagenet-1K database (Deng et al., 2009). We use $128 \times 128$ crops (formed by scaling the smaller side to 128, and taking a random crop along the other dimension) of 160k images from a subset of Imagenet classes (ids 152 to 281) of different canines. We use similar settings as for faces, but feed a higher 200-dimensional noise vector to the generator, which also begins by mapping this to a feature vector that is twice as large (2048), and which has an extra transpose-convolution layer to go upto the $128 \times 128$ resolution. We again use $8 \times 8$ convolutional filters with stride two to form $\{W_k^T\}$—in this case, these produce $64 \times 64$ single channel images. We use only $K = 12$ discriminators, each of which has an additional layer because of the higher-resolution—beginning with fewer channels in the first layer so that the final feature vector used for classification is the same length as for faces. Figure 6 shows manually selected samples after 100k iterations of training (see supplementary material for a larger random set). We see that since it is trained on a more diverse and unaligned image content, the generated images are not globally plausible photographs. Nevertheless, we find that the produced images are sharp, and that generator learns to reproduce realistic low-level textures as well as some high-level composition.

## 5 Conclusion

In this paper, we proposed a new framework to training GANs for high-dimensional outputs. Our approach employs multiple discriminators on random low-dimensional projections of the data to stabilize training, with enough projections to ensure that the generator learns the true data distribution. Experimental results demonstrate that this approaches leads to more stable training, with generators continuing to improve for longer to ultimately produce higher-quality samples. Source code and trained models for our implementation is available at the project page [**anonymized for review**].

In our current framework, the number of discriminators is limited by computational cost. In future work, we plan to investigate training with a much larger set of discriminators, employing only a small subset of them at each iteration, or every set of iterations. We are also interested in using multiple discriminators with modified and regularized objectives (*e.g.*, (Nowozin et al., 2016; Arjovsky et al., 2017; Zhao et al., 2016)). Such modifications are complementary to our approach, and deploying them together will likely be beneficial.

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

# Supplementary Material

## A  THEORY

In this section we provide additional theoretical results showing the benefits of random projections, for some particular choice of distributions, for both improving stability and guaranteeing consistency of GAN training.

### STABILITY

Here, we consider several simplifying assumptions on the distribution of $x$, to gain intuition as to how a random low-dimensional projection affects the "relative" volume of the support of $\mathbb{P}_x$. Let us assume that the range of $x$ is the $d$-dimensional ball $B^d$ of radius 1 centered at 0. We define the support $supp(\mathbb{P}_x) \subset B^d$ of a distribution $\mathbb{P}_x$ to be the set where the density is greater than some small threshold $\epsilon$. Assume that the projection $W \in \mathbb{R}^{d \times m}$ is entry-wise random Gaussian (rather than corresponding to a random convolution). Denote as $B_W^d$ the projection of the range on $W$, and $\mathbb{P}_{W^T x}$ as the marginal of $\mathbb{P}_x$ along $W$.

**Theorem A.1.** *Assume* $\mathbb{P}_x = \sum_j \tau_j \mathcal{N}(x | \mu_j, \Sigma_j)$ *is a mixture of Gaussians, such that the individual components are sufficiently well separated (in the sense that there is no overlap between their supports or the projections thereof, see (Dasgupta, 1999)). If* $supp(\mathbb{P}_x) \subset B^d$ *and* $Vol(supp(\mathbb{P}_x)) > 0$, *then* $Vol(supp(\mathbb{P}_{W^T x})) / Vol(B_W^d) > Vol(supp(\mathbb{P}_x)) / Vol(B^d)$ *with high probability.*

This result implies that the projection of the support of $\mathbb{P}_x$, under this simplified assumptions, occupies a higher fraction of the volume of the projection of the range of $x$. This aids in stability because it makes it more likely that a larger fraction of the generator's samples (which also lie within the range of $x$) will, after projection, overlap with the projected support $supp(\mathbb{P}_x)$, and can not be rejected absolutely by the discriminator.

**Proof of Theorem A.1.**   We first show that we can assume that the columns of the projection $W$ are orthonormal. Since $W \in \mathbb{R}^{d \times m}$ is entry-wise Gaussian distributed, it has rank $m$ with high probability. Then, there exists a square invertible matrix $A$ such that $W' = AW$ where $W'$ is orthonormal. In that case, $Vol(supp(\mathbb{P}_{W^T x})) / Vol(B_W^d) = Vol(supp(\mathbb{P}_{W'^T x})) / Vol(B_{W'}^d)$ because the numerator and denominator terms for both can be related by $\det(A)$ for the change of variables, which cancels out. Note that under this orthonormal assumption, $B_W^d = B^m$.

Next, we consider the case of an individual Gaussian distribution $\mathbb{P}_x = \mathcal{N}(x | \mu, \Sigma)$, and prove that the ratio of supports (defined with respect to a threshold $\epsilon$) does not decrease with the projection. The expression for these ratios is given by:

$$Vol(supp(\mathbb{P}_x)) = Vol(B^d) \times \det(\Sigma) \times \left[ \log \frac{1}{\epsilon^2} - d \log 2\pi - \log \det(\Sigma) \right]$$

$$\Rightarrow \frac{Vol(supp(\mathbb{P}_x))}{Vol(B^d)} = \det(\Sigma) \times \left[ \log \frac{1}{\epsilon^2} - d \log 2\pi - \log \det(\Sigma) \right]. \tag{4}$$

$$\frac{Vol(supp(\mathbb{P}_{W^T x}))}{Vol(B_W^d)} = \det(W^T \Sigma W) \times \left[ \log \frac{1}{\epsilon^2} - m \log 2\pi - \log \det(W^T \Sigma W) \right]. \tag{5}$$

For sufficiently small $\epsilon$, the volume ratio of a single Gaussian will increase with projection if $\det(W^T \Sigma W) > \det(\Sigma)$. Note that all eigenvalues of $\Sigma \leq 1$, with at-least one eigenvalue strictly $< 1$ (since $supp(\mathbb{P}_x) \subset B^d$). First, we consider the case when $\Sigma$ is not strictly positive definite and one of the eigenvalues is 0. Then, $Vol(supp(\mathbb{P}_x)) = 0$ and $Vol(supp(\mathbb{P}_{W^T x})) \geq 0$, *i.e.,* the volume ratio either stays the same or increases.

For the case when all eigenvalues are strictly positive, consider a co-ordinate transform where the first $m$ co-ordinates of $x$ correspond to the column vectors of $W$, such that

$$\Sigma = \left[ \begin{array}{cc} \Sigma_W & \Sigma_{WW'} \\ \Sigma_{WW'}^T & \Sigma_{W'} \end{array} \right], \tag{6}$$

where $\Sigma_W = W^T \Sigma W$. Then,

$$\begin{aligned} \det(\Sigma) &= \det(\Sigma_W) \det(\Sigma_{W'} - \Sigma_{WW'}^T \Sigma_W^{-1} \Sigma_{WW'}) \\ &\leq \det(\Sigma_W) \det(\Sigma_{W'}), \\ \Rightarrow \det(\Sigma_W) &\geq \det(\Sigma) / \det(\Sigma_{W'}). \end{aligned} \tag{7}$$

Note that $\det(\Sigma_{W'}) \leq 1$, since all eigenvalues of $\Sigma$ are $\leq 1$, with equality only when $W$ is completely orthogonal to the single eigenvector whose eigenvalue is strictly $< 1$, which has probability zero under the distribution for $W$. So, we have that $\det(\Sigma_{W'}) < 1$, and

$$\det(W^T \Sigma W) = \det(\Sigma_w) > \det(\Sigma). \tag{8}$$

The above result shows that the volume ratio of individual components never decrease, and *always* increase when their co-variance matrices are full rank (no zero eigenvalue). Now, we consider the case of the Gaussian mixture. Note that the volume ratio of the mixture equals the sum of the ratios of individual components, since the denominator $\text{Vol}(B^m)$ is the same, where the support volume in these ratios for component $j$ is defined with respect to a threshold $\epsilon/\tau_j$. Also, note that since mixture distribution has non-zero volume, at least one of the Gaussian components must have all non-zero eigenvalues. So, the volume ratios of $\mathbb{P}_x$ and $\mathbb{P}_{W^T x}$ are both sums of individual Gaussian component terms, and each term for $\mathbb{P}_{W^T x}$ is greater than or equal to the corresponding term for $\mathbb{P}_x$, and at least one term is strictly greater. Thus, the support volume ratio of $\mathbb{P}_{W^T x}$ is strictly greater than that of $\mathbb{P}_x$.

CONSISTENCY

**Proof of Theorem 3.1.**   The proof follows along the same steps as that of Theorem 1 in Goodfellow et al. (2014).

$$\begin{aligned} V(D_k, G) &\\ &= \mathbb{E}_{x \sim \mathbb{P}_x}[\log D_k(W_k^T x)] + \mathbb{E}_{x \sim \mathbb{P}_g}[\log(1 - D_k(W_k^T x))] \\ &= \mathbb{E}_{Y \sim \mathbb{P}_{W_k^T x}}[\log D_k(y)] + \mathbb{E}_{y \sim \mathbb{P}_{W_k^T g}}[\log(1 - D_k(y))]. \end{aligned} \tag{9}$$

For any point $y \in \text{supp}(\mathbb{P}_{W_k^T x}) \cup \text{supp}(\mathbb{P}_{W_k^T g})$, differentiating $V(D_k, G)$ w.r.t. $D_k$ and setting to 0 gives us:

$$D_k(y) = \frac{\mathbb{P}_{W_k^T x}(y)}{\mathbb{P}_{W_k^T x}(y) + \mathbb{P}_{W_k^T g}(y)}. \tag{10}$$

Notice we can rewrite $V(D_k, G)$ as

$$V(D_k, G) = -2\log(2) + KL\left(\mathbb{P}_{W_k^T x} || \frac{\mathbb{P}_{W_k^T x} + \mathbb{P}_{W_k^T g}}{2}\right)$$
$$+ KL\left(\mathbb{P}_{W_k^T g} || \frac{\mathbb{P}_{W_k^T x} + \mathbb{P}_{W_k^T g}}{2}\right). \tag{11}$$

Here KL is the Kullback Leibler divergence, and it is easy to see that the above expression achieves the minimum value when $\mathbb{P}_{W_k^T x} = \mathbb{P}_{W_k^T g}$.

Next, we present a result that shows that given enough random projections $K$, the full distribution $\mathbb{P}_g$ of generated samples will closely match the full true distribution $\mathbb{P}_x$.

**Def:** A function $f : \mathbb{R}^d \to \mathbb{R}_+$ is $L$-Lipschitz, if $\forall\, x_1, x_2 \in \mathbb{R}^d, |f(x_1) - f(x_2)| \leq L \cdot d(x_1, x_2)$.

**Theorem A.2.** *Let $\mathbb{P}_x$ and $\mathbb{P}_g$ be two compact distributions with support of radius $B$, such that $\mathbb{P}_{W_k^T x} = \mathbb{P}_{W_k^T g}, \forall k \in \{1, \cdots, K\}$. Let $\{W_k\}$ be entrywise random Gaussian matrices in $\mathbb{R}^{d \times m}$. Let $R = \mathbb{P}_x - \mathbb{P}_g$ be the residual function of the difference of densities. Then, with high probability, for any $x \in \mathbb{R}^d$,*

$$|R(x)| = |\mathbb{P}_x(x) - \mathbb{P}_g(x)| \leq O\left(\frac{BL_R}{K^{\frac{1}{d-m}}}\right),$$

*where $L_R$ is the Lipschitz constant of $R$.*

This theorem captures how much two probability densities can differ if they match along $K$ marginals of dimension $m$, and how the error decays with increasing number of discriminators $K$. In particular, if we have a smooth residual function $R$ (with small Lipschitz constant $L_R$), then at any point $\in \mathbb{R}^d$, it is constrained to have small values—smaller with increasing number of discriminators $K$, and higher dimension $m$. The dependence on $L_R$ suggests that the residual can take larger values if it is not smooth—which can result from either the true density $\mathbb{P}_x$ or the generator density $\mathbb{P}_g$, or both, being not smooth.

Again, this result gives us rough intuition about how we may expect the error between the true and generator distribution to change with changes to the values of $K$ and $m$. In practice, the empirical performance will depend on the nature of the true data distribution, the parametric form / network architecture for the generator, and the ability of the optimization algorithm to get near the true optimum in Theorem 3.1.

**Proof of Theorem A.2.** Let $R = \mathbb{P}_x - \mathbb{P}_g$, be the residual function that captures the difference between the two distributions. $R$ satisfies the following properties:

$$\int_x R(x)dx = \int_x (\mathbb{P}_x(x) - \mathbb{P}_g(x))dx = 1 - 1 = 0, \tag{12}$$

and for any set $S$,

$$\int_{x \in S} R(x)dx \leq \int_{x \in S} \mathbb{P}_x(x)dx \leq 1. \tag{13}$$

Further, since both the distributions have same marginals along $k$ different directions $\{W_k\}$, we have, for any $x$,

$$R_{W_k^T y}(x) = \int_{y|x = W_k^T y} R(y) = 0.$$

We first prove this result for discrete distributions supported on a compact set $\mathcal{S}$ with $\gamma$ points along each dimension. Let $\tilde{\mathbb{P}}$ denote such a discretization of a distribution $\mathbb{P}$ and $\tilde{R} = \tilde{\mathbb{P}}_x - \tilde{\mathbb{P}}_g$.

Each of the marginal equation $\tilde{\mathbb{P}}_{W_k^T x} = \tilde{\mathbb{P}}_{W_k^T g}$ $(\tilde{R}_{W_k^T y} = 0)$ is equivalent to $\gamma^m$ linear equations of the distribution $\tilde{\mathbb{P}}_x$ of the form, $\sum_{x:W_k^T x = y} \tilde{\mathbb{P}}_x(x) = \tilde{\mathbb{P}}_{W_k^T g}(y)$. Note that we have $\gamma^d$ choices for $x$ and $\gamma^m$ choices for $y$. Let $A_k \in \mathbb{R}^{\gamma^m \times \gamma^d}$ denote the coefficient matrix $A_k \tilde{\mathbb{P}}_x = \tilde{\mathbb{P}}_{W_k^T g}$, such that $A_k(i,j) = 1$ if $W_k^T x_i = y_j$, and 0 otherwise.

The rows of $A_k$ for different values of $y_j$ are clearly orthogonal. Further, since different $W_k$ are independent Gaussian matrices, rows of $A_k$ corresponding to different $W_k$ are linearly independent. In particular let $A \in \mathbb{R}^{\gamma^m K \times \gamma^d}$ denote the vertical concatenation of $A_k$. Then, $A$ has full row rank of $\gamma^m \cdot K$ with probability $\geq 1 - c \cdot m \cdot e^{-d}$ (Vershynin, 2010), with $c$ some arbitrary positive constant. Since $\tilde{\mathbb{P}}_x$ is a $\gamma^d$ dimensional vector, $\gamma^d$ linearly independent equations determine it uniquely. Hence $\gamma^m \cdot K \geq \gamma^d$, guarantees that $\tilde{\mathbb{P}}_x = \tilde{\mathbb{P}}_g$ or $\tilde{R} = 0$.

Now we extend the results to the continuous setting. Without loss of generality, let the compact support $\mathcal{S}$ of the distributions be contained in a sphere of radius $B$. Let $\mathcal{N}_{\frac{\epsilon}{L_R}}$ be an $\frac{\epsilon}{L_R}$ net of $\mathcal{S}$, with $\gamma^d$ points (see Lemma 5.2 in Vershynin (2010)), where $\gamma = 2B \cdot L_R/\epsilon$. Then for every point $x_1 \in \mathcal{S}$, there exists a $x_2 \in \mathcal{N}_{\frac{\epsilon}{L_R}}$ such that, $d(x_1, x_2) \leq \frac{\epsilon}{L_R}$.

Further for any $x_1, x_2$ with $d(x_1, x_2) \leq \frac{\epsilon}{L_R}$, if $R$ is $L_R$ Lipschitz we know that,

$$|R(x_1) - R(x_2)| \leq L_R \cdot \frac{\epsilon}{L_R} = \epsilon. \tag{14}$$

Finally, notice that the marginal constraints do not guarantee that the distributions $\tilde{\mathbb{P}}_{W_k^T x}$ and $\tilde{\mathbb{P}}_{W_k^T g}$ match exactly on the $\epsilon$-net (or that $R$ is 0), but only that they are equal upto an additive factor of $\epsilon$. Hence, combining this with equation 14 we get, $|R(x)| = |\mathbb{P}_x(x) - \mathbb{P}_g(x)| \leq O(\epsilon)$, for any $x$ with probability $\geq 1 - c \cdot m \cdot K \cdot e^{-d}$. Since we have $\gamma^{d-m} = \left(\frac{BL}{\epsilon}\right)^{d-m} = O(\frac{1}{K})$, we get $\epsilon = O(\frac{BL}{K^{\frac{1}{d-m}}})$.

# B  ADDITIONAL EXPERIMENTAL RESULTS

**Face Images: Proposed Method ($K = 48$)**

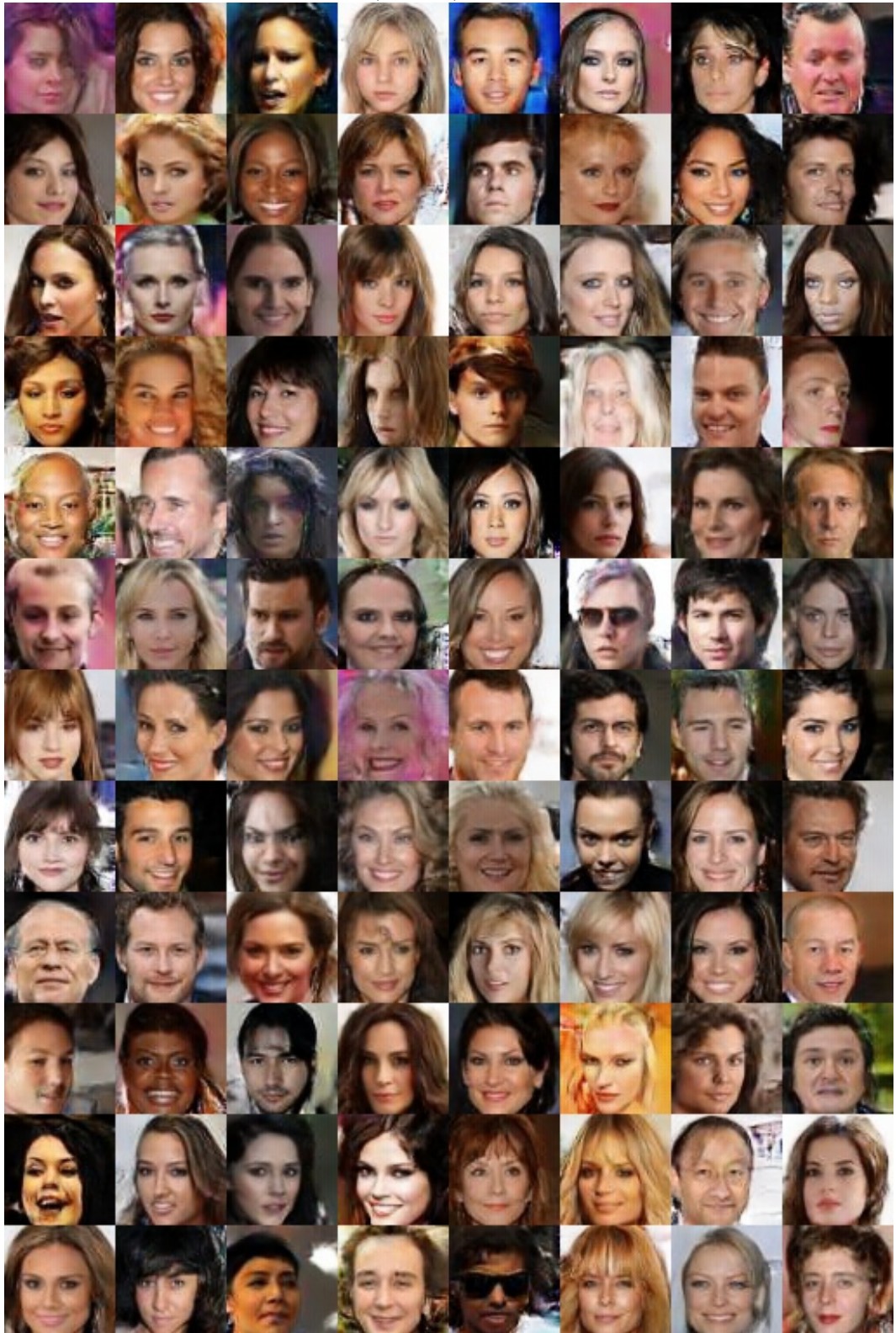

**Face Images: Proposed Method** ($K = 24$)

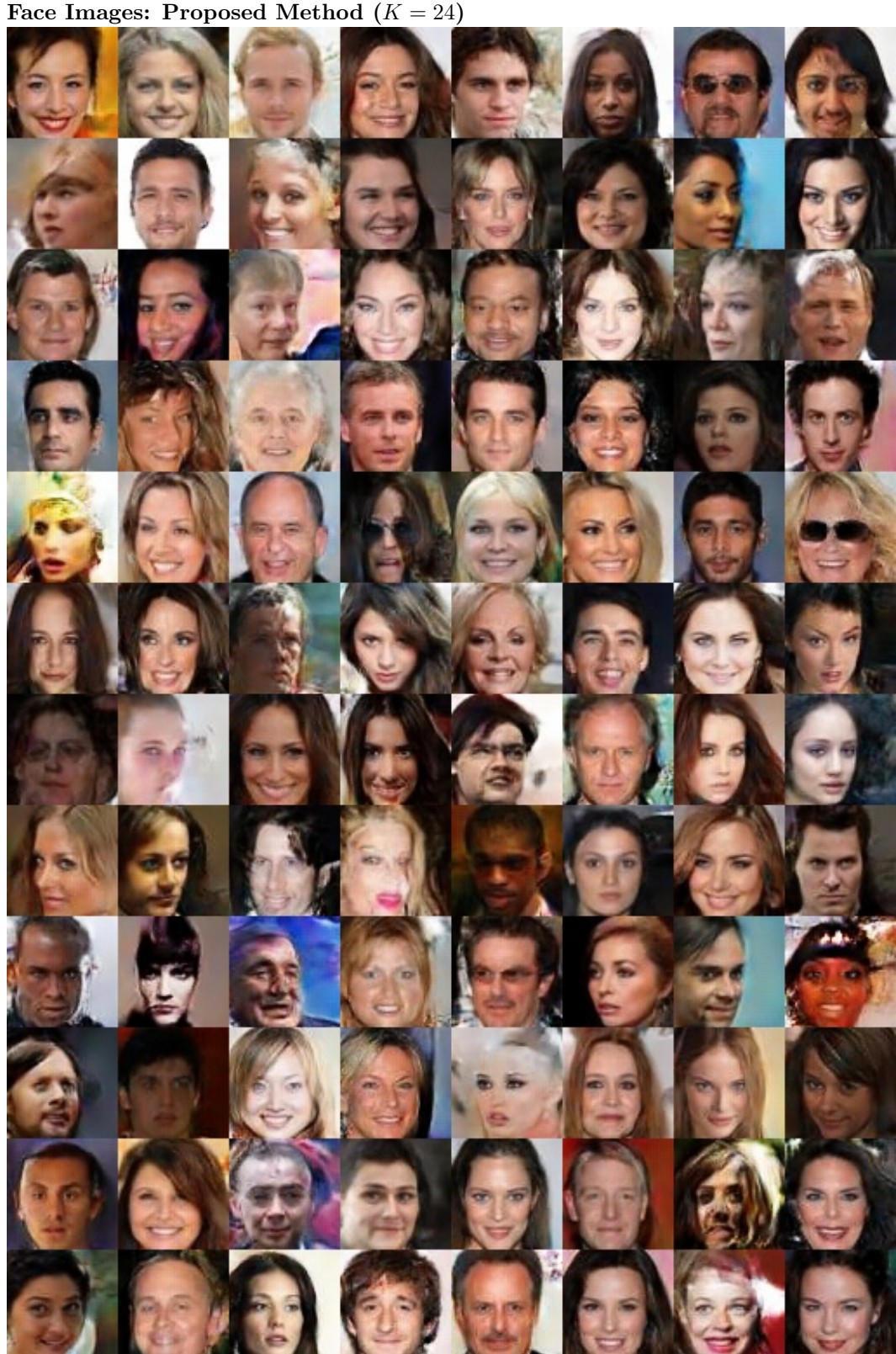

**Face Images: Proposed Method** ($K = 12$)

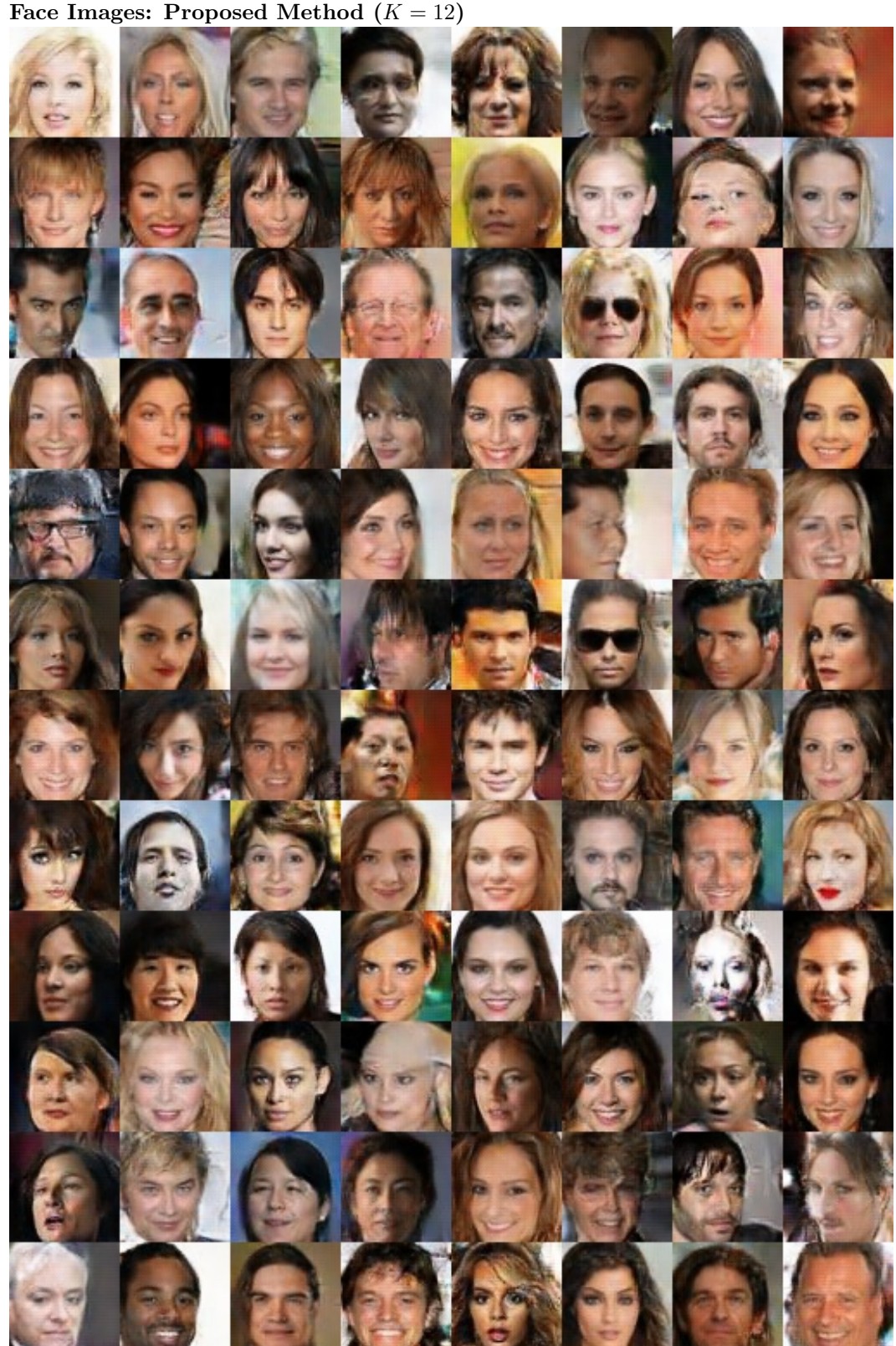

**Face Images: Traditional DC-GAN (Iter. 40k)**

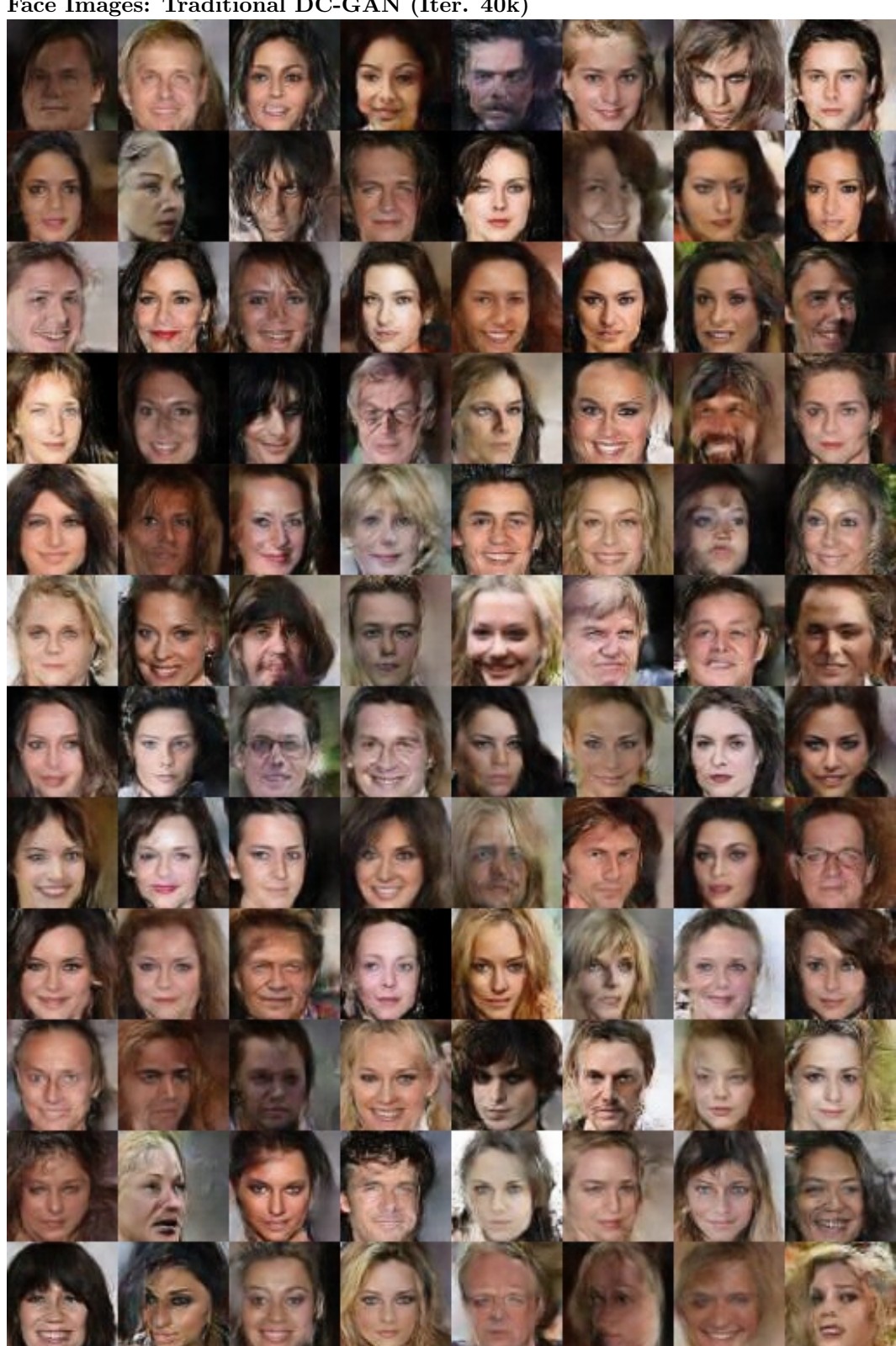

**Face Images: Traditional DC-GAN (Iter. 100k)**

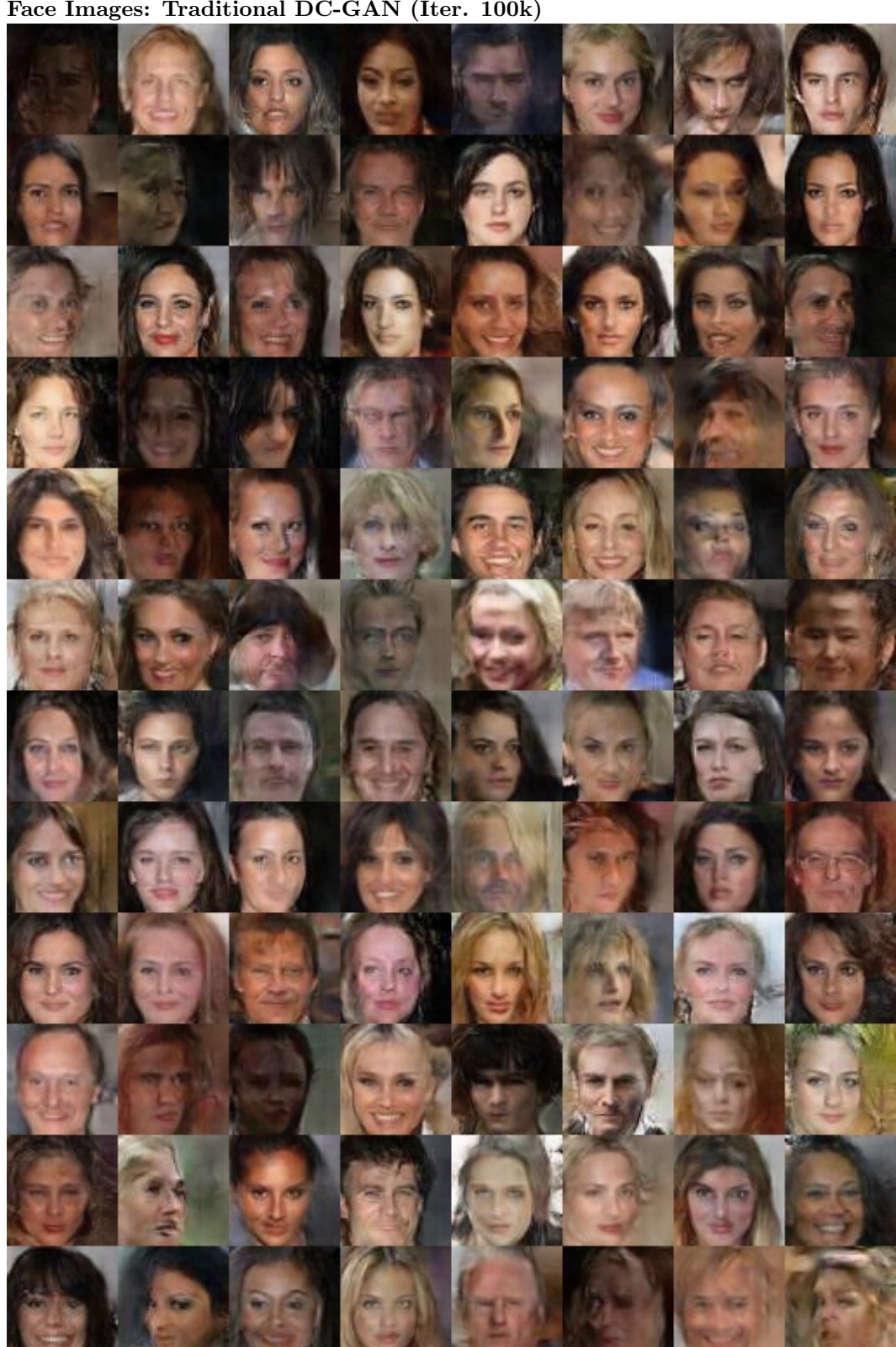

**Random Imagenet-Canine Images: Proposed Method**

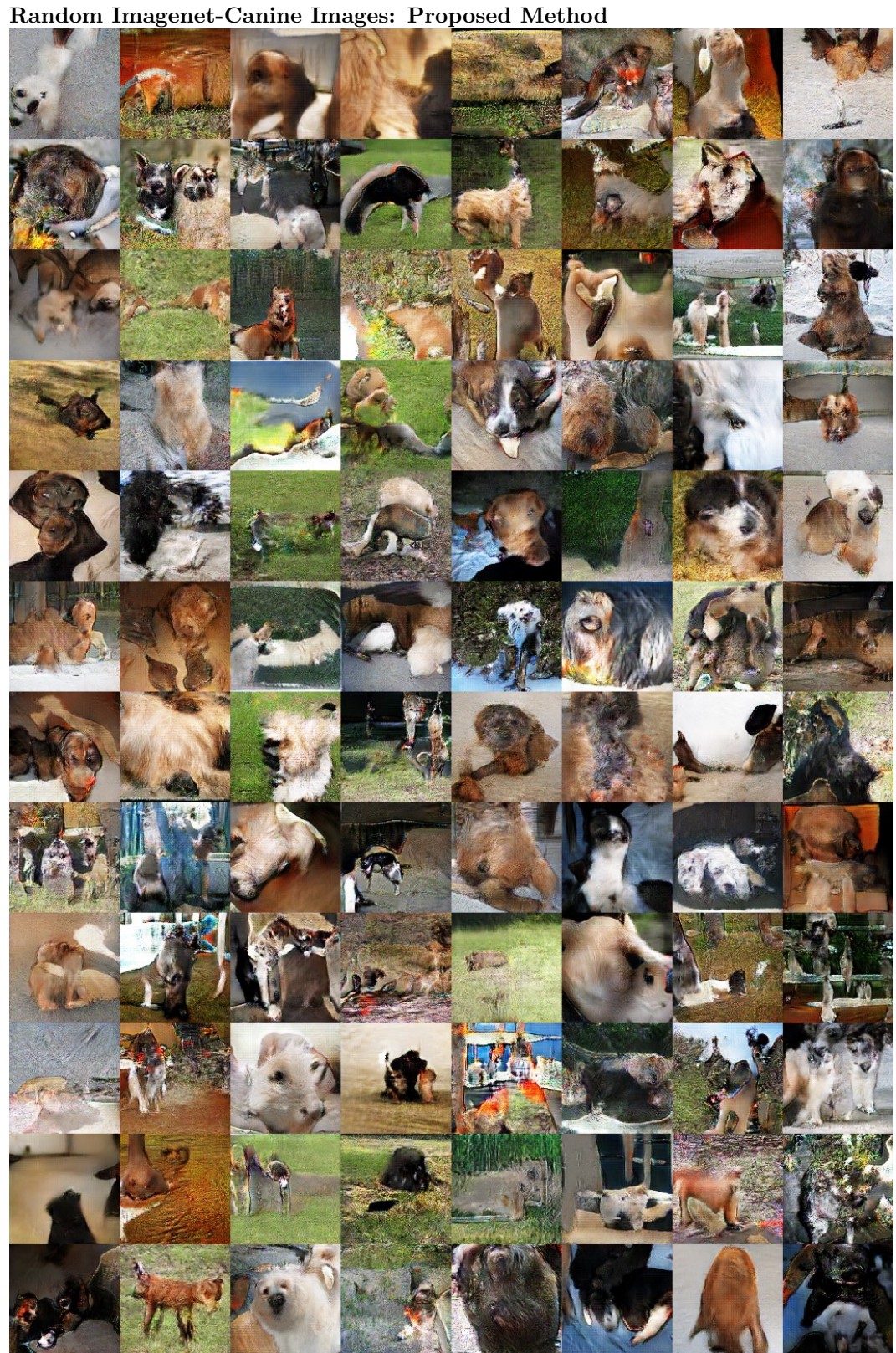

