# OpenReview forum: "Stabilizing GAN Training with Multiple Random Projections"
_ICLR.cc/2018/Conference — Reject_

### Official Review · AnonReviewer2 · 2017-11-26
**Review of Stabilizing GAN Training with Multiple Random Projections**

**Rating:** 5
**Confidence:** 4

**Review:**


The paper proposes to stabilize GAN training by using an ensemble of discriminators, each workin on a random projection of the input data, to provide the training signal for the generator model.

Q1: “In relation to “Theorem 3.1. … will produce samples from a distribution whose marginals along each of the projections W_k match those of the true distribution”.. I presume an infinite number of generator distributions could give rise to the correct marginals however not necessarily be converged to the data distribution. In Theorem A.2 the authors upperbound this residual as a function of the smoothness and support of the distributions as well as the projections presented to the discriminators. Can the authors comment on how tight this bound is e.g. as a function the number of used discriminators or the choosen projection methods ?

Q2: Related to the above. Did the authors do or considered any frequency analysis of the ensemble of random projection? I guess you could easily do a numeric simulation of the expected frequency spectrum of the combined set discriminators?


Q3: My primary concern with the work is the above mentioned computational complexity of running K discriminators in parallel. This is especially in relation to the experimental results showing significant high-frequency artefacts when running with K=12 classifiers (K=12 celebA results and “Random Imagenet-Canine Images: Proposed Method” in suplementary results). I think this is as expected as the authors are effectively fitting each classifier to the distributions of smoothed  (with 8x8 random kernel)  subsampled version of the input image. I would expect that each discriminator sees none or only a very limited amount  the high frequency component in the images.  Do the authors have any comments on how the sampling of the projection kernels affects the image results especially if the number of needed classifiers can be reduced somehow? I would expect that a combination of smoothing and high frequency filters would be needed to remove the high frequency artefacts?

Q4: Whats the explanation of the oscilating patterns in figure 2?

Q5: In the conclusion the authors mention that their framework is currently limited by the computational of running K discriminators and proposes:

“In our current framework, the number of discriminators is limited by computational cost. In future work, we plan to investigate training with a much larger set of discriminators, employing only a small subset of them at each iteration, or every set of iterations”

In the extreme case of only using a single randomly discriminator the approach is quite similar to the quite widely used input dropout to the discriminator?

Overall I like the simplicity of the proposed idea. However i’m not completely convinced that the “marginal” convergence proof holds for the relative low number of discriminators possible to use in practice. At least i would like the authors to touch on this key aspect of the method both theoretically and with experiments/simulations. Also several other methods have recently been proposed to improve stability of GANs, however no experimental comparisons is made with these methods (WGAN, EGAN, LSGAN etc.)

---

> ### Author Response · Authors · 2017-12-18
> **Response**
>
>
> We thank the reviewer for their comments and detailed observations. We address these individually below:
>
> Q1: Without further assumptions on the data, the bound is tight in the problem parameters. But, as the reviewer also notes, there usually is additional structure (conditional independence,  sparsity, etc.) in natural data distributions that makes the approach succeed with fewer projections. We're interested in exploring more precise characterizations, but this would have to be domain / data specific. The proof and analysis technique used for Thm A.2 will serve as a useful starting point for such domain-specific analysis: still using that each discriminator constrains a different marginal, we hope to exploit assumptions about structure to provide tighter bounds on the error as a function of the number of such marginal constraints.
>
> While domain-specific analysis is an important direction of future work, we believe it is beyond the scope of this paper, which we see as making the first step in introducing the simple idea of using random projections ensembles to stabilize GAN training, showing that it has promise and utility, and in providing a starting point for analyzing this setup. We believe that the general idea will be applicable over a broad range of domains (data types, conditional vs general GANs, etc.), perhaps each with their own adaptations and extensions, and we believe that the paper as-is will therefore be of interest to a broader community.
>
> Q2: Each projection is a combination of filtering + downsampling. Since the filters are random iid, in terms of spatial frequency (ignoring color), they have a flat expected power spectral density---in other words, they're as likely to be high-pass as low-pass. Downsampling then folds in the higher freq. quarters of the spectrum to produce an aliased image. Fundamentally, the projection operation (again, ignoring the projection along color channels) can be seen as taking sets of frequency components, and retaining different random linear combinations of these sets in each projection.
>
> The noise one sees in the K=12 case is actually "periodic" noise, with a period equal to the sampling rate (2), and can affect low and high-frequencies equally. Basically, with too few discriminators, the generator can choose to generate the 'right' weighted sum of each of the aliased low and high frequency sets. With enough discriminators which look at different weighted aliased combinations, the generator is more and more constrained in getting each element of that set right individually.
>
> Q3: The reviewer's intuition is correct---the generator does fit the aliased version (rather than the smoothed version) of the data when trained against too few discriminators. This also follows from the intuition from the proof of Theorem A.2. Regarding choosing efficient projections, this would again have to depend on the data distribution/be domain specific.
>
> Actually, one approach we're actively pursuing currently (as follow-on work) applies to the domain of natural images is related to the reviewer’s comments---we're exploring a multi-resolution approach with crops over a wavelet pyramid. This approach is motivated through a modeling assumption of conditional independence of coefficients in the pyramid (two fine-level coefficients are independent conditioned on scaling coefficients).
>
> But note that while domain-specific efficient projections are desirable, random projections will succeed with a large enough number of discriminators (and may be the only choice in some applications). To that end, we want to highlight that the issue of computation cost is mitigated to some extent by the fact that the forward/backward pass through the multiple discriminators can be done in parallel (on multiple GPUs). This means that for applications where a large number of discriminators is required under the proposed approach, one could still train the generator in the same amount of time given access to more computational resources. And at best, it will provide a starting point for searching for more efficient projections.
>
> Q4: These are due to orbits between the discriminator/generator---discriminator improves, then generator catches up, etc. (we see this with a single discriminator as well).
>
> Q5: This would be different from dropout because we'd still have a different discriminator for each 'drop configuration'. The idea would be to keep the discriminators for every projection around, but only train / use-for-generator-update a few or one of them in each iteration. But one can think of this as 'dropping' parts of the loss term for the generator.
>
> - Finally, note our approach is complementary to the other methods mentioned by the reviewer. Our approach addresses the 'high-dimensional' aspect of the stability problem. But it can be used with better losses (like WGAN), better architectures, etc, because one can apply the approach of operating on multiple random projections to such versions.

---

### Official Review · AnonReviewer3 · 2017-11-27
**Due to poor experimental validation and inconclusive results, the reviewer does not recommend acceptance of the paper.**

**Rating:** 3
**Confidence:** 5

**Review:**


- Paper summary

The paper proposes a GAN training method for improving the training stability. The key idea is to let a GAN generator competes with multiple GAN discriminators where each discriminator takes a random low-dimensional projection of an input image for differentiate whether the input image is a real or generated one. Visual generation results from the proposed method with comparison to those generated by the DCGAN were used as the main experimental validation for the merit of the proposed method. Due to poor experimental validation and inconclusive results, the reviewer does not recommend the acceptance of the paper.

- Inconclusive results

The paper fails to compare the proposed method with the GMAN framework [a], which was the first work proposing utilizing multiple discriminators for more stable GAN training. Without comparing to the GMAN work, we do not know whether the benefit is from using multiple discriminators proposed in the GMAN work or from using the random low dimensional projections proposed in this paper. If it is former, then the proposed method has no merits at all.

In addition, the generator loss curve shown in Figure 2 is not making much sense. The generator loss curve will be meaningful if each discriminator update is optimal. However, this is not the case in the proposed method. There is little to conclude from Figure 2.

[a] Durugkar et al. "Generative multi-adversarial networks." ICLR 2017

- Poor experimental validation

The paper fails to utilize more established performance metrics such as the inception loss or human evaluation score to evaluate its benefit. It does not compare to other approaches for stabilizing GAN training such as WGAN or LSGAN. The main results shown in the paper are generating 64x64 human face images, which is not impressive.

---

> ### Author Response · Authors · 2017-12-18
> **Response**
>
>
> We'd like to clarify that our paper is focused on the specific goal of addressing instability in training GANs in high-dimensions. (Arjovsky et al. provide an excellent description of this phenomenon as well as more context to the problem we're trying to solve). Our theoretical analyses and experimental evaluations are therefore both geared towards this goal. We respond to specific questions below, but would like to respectfully ask the reviewer to take a second look at the paper in this context (rather than the generic context of improved GAN results) to see if it changes their mind.
>
> - We compare to DCGAN in order to fix a reasonably successful yet generic architecture, and then to isolate the effect of training with a single full-dimensional discriminator, and multiple low-dimensional discriminators. There are definitely better architectures and loss functions (e.g., WGAN) out there, but ours is a training approach that can be applied with those architectures and losses as well.
>
> - Figure 2 shows that the low-dimensional discriminators don't saturate like they do with a high-dimensional one. The curves in Figure 2 are not meant to analyze quality of the discriminator (and hence don't make sense in that context), but they do show that the discriminator isn't able to perfectly separate real and fake samples (at which point, as described in Arjovsky et al., its gradients become meaningless and this causes training to diverge). It is Figure 3 that compares quality by showing generated faces across training.
>
> - Note that we cite the Durugkar et al. paper and  GMAN method, and discuss it in Sec 1.1, along with a host of other ensemble approaches. But again, the goal of GMAN is different, and is to better approximate the optimization over the discriminator. They do not address stability (like all other methods for GAN training, they stop training early).  Further note the fact that our experiments already show that a single full-dimensional discriminator will saturate. With the goal of evaluating stability, using multiple full dimensional discriminators would not be useful since they would all also saturate (they have the same capacity, and if anything, they have an even higher advantage over the generator).

---

### Official Review · AnonReviewer1 · 2017-11-28
**Stabilizing GAN Training with Multiple Random Projections**

**Rating:** 8
**Confidence:** 4

**Review:**

The paper proposes a new approach to GAN training whereby they train one generator against an ensemble of discriminative that each receive a randomly projected version of the data. The authors show that this approach provides stable gradients to train the generator.

This is a nice idea, and both the theoretical analysis presented and the experiments on image data sets are interesting. Although the idea to train an ensemble of learning machines is not new, see e,.g. [1,2] -- and it would be useful to add some background on this, and the regularisation effect that emerges from it --  it does become new in the new context considered here, as the paper shows that such ensemble can also fulfil the role of stabilising GAN training.
The results are quite convincing that the proposed method is useful in practice,

It would be interesting to know if weighting the discriminators, or discarding the unlucky random projections as it was done in [1] would have potential in this context?

[1] Timothy I. Cannings, Richard J. Samworth. Random-projection ensemble classification. Journal of the Royal Statistical Society B, 79(4), 2017, Pages 959-1035.
[2] Robert J. Durrant, Ata Kabán. Random projections as regularizers: learning a linear discriminant from fewer observations than dimensions. Machine Learning 99(2), 2015, Pages 257-286.

---

> ### Author Response · Authors · 2017-12-18
> **Response**
>
> - We thank the reviewer for their encouraging comments as well as for pointers to [1,2]. Using ensembles as well as random projections has a rich history as means to solve a variety of challenges in machine learning. Our work is motivated by these successes, and aims to show that these ideas are useful also for the challenge of instability in training GANs in high dimensions.
>
> We draw connections to some prior works in Sec 1.1, and adding these works to the discussion (especially [1]) would definitely make it more informative. Thanks !
>
> - Discarding discriminators / projections is an interesting idea ! To some degree, the unlucky discriminators are already being weighted down because as they saturate, their contribution to the sum of gradients from all discriminators already goes to 0. But it would be interesting to see if we can make training more 'efficient', by discarding saturated discriminators and restarting them with a different projection matrix. Combining this idea with some of the ones we discuss in the conclusion is definitely an interesting direction of future work, and one that we intend to explore.

---

### Author Response · Authors · 2018-01-05
**Revision**

We've posted a revision adding the two papers suggested by R1 to the related work section. Individual responses to the reviews were posted below earlier.

---

### Decision · Program_Chairs · 2018-01-29
**ICLR 2018 Conference Acceptance Decision**

**Decision:**

Reject

**Comment:**

The paper proposes to use multiple discriminators to stabilize the GAN training process. Additionally, the discriminators only see randomly projected real and generated samples.

Some valid concerns raised by the reviewers which makes the paper weak:
 - Multiple discriminators have been tried before and the authors do not clearly show experimentally / theoretically if the random projection is adding any value.
- Authors compare only with DCGAN and the results are mostly subjective. How much improvement the proposed approach provides when compared to other GAN models that are developed with stability as the main goal is hence not clear.